# Differentially Private Selection from Secure Distributed Computing

## ABSTRACT

Given a collection of vectors $\boldsymbol{x}^{(1)}, \ldots, \boldsymbol{x}^{(n)} \in \{0,1\}^d$, the *selection* problem asks to report the index of an "approximately largest" entry in $\boldsymbol{x} = \sum_{j=1}^{n} \boldsymbol{x}^{(j)}$. Selection abstracts a host of problems, for example: Recommendation of a popular item based on user feedback; releasing statistics on the most popular web sites; hyperparameter tuning and feature selection in machine learning. We study selection under differential privacy, where a released index guarantees privacy for individual vectors. Though selection can be solved with an excellent utility guarantee in the central model of differential privacy, the distributed setting where no single entity is trusted to aggregate the data lacks solutions. Specifically, strong privacy guarantees with high utility are offered in high trust settings, but not in low trust settings. For example, in the popular *shuffle model* of distributed differential privacy, there are strong lower bounds suggesting that the utility of the central model cannot be obtained. In this paper we design a protocol for differentially private selection in a trust setting similar to the shuffle model—with the crucial difference that our protocol tolerates corrupted servers while maintaining privacy. Our protocol uses techniques from secure multi-party computation (MPC) to implement a protocol that: (i) has utility on par with the best mechanisms in the central model, (ii) scales to large, distributed collections of high-dimensional vectors, and (iii) uses $k \geq 3$ servers that collaborate to compute the result, where the differential privacy guarantee holds assuming an honest majority. Since general-purpose MPC techniques are not sufficiently scalable, we propose a novel application of *integer secret sharing*, and evaluate the utility and efficiency of our protocol both theoretically and empirically. Our protocol improves on previous work by Champion, shelat and Ullman (CCS '19) by significantly reducing the communication costs, demonstrating that large-scale differentially private selection with information-theoretical guarantees is feasible in a distributed setting.

### ACM Reference Format:
Anonymous Author(s). 2023. Differentially Private Selection from Secure Distributed Computing. In *Proceedings of ACM Conference (Conference'17)*. ACM, New York, NY, USA, 13 pages. https://doi.org/10.1145/nnnnnnn.nnnnnnn

## 1 INTRODUCTION

Differentialy private *selection* of the largest entry in a vector enables data analysis on sensitive datasets—for example announcing the winning candidate in a vote, or identifying a common genetic marker from a set of DNA sequences. While there exist solutions to the selection problem with strong guarantees scaling logarithmically with dimension and independent of the size of vector entries (e.g., [MS20]), they operate in the *central model* of differential privacy, which requires trust in a single party to perform the computation. Existing solutions with weaker trust assumptions, on the other hand, scale poorly or require significantly more noise to maintain privacy.

A pragmatic solution is to aim for a middle ground: distributing trust among multiple parties. This setting is natural when a person trusts a party (e.g., their local hospital) with their data, but not *every* party (e.g., they may not want to share their data with every hospital). In principle, every mechanism in the central model of differential privacy could be simulated in such a distributed setting using techniques for secure multi-party computation (MPC), but that approach is not viable in general because MPC is not yet practical for large-scale general-purpose computations. Steinke [Ste20] introduced a more restricted class of protocols working in the so-called *multi-central model*, in which data holders submit information to $k$ servers, which then communicate and compute the output of the mechanism. An attractive property of this model is that data holders only need to submit a single message to each server, after which no involvement is needed. Nevertheless, techniques such as additive secret sharing allow protocols that have high utility and protect privacy even if $k-1$ servers share their information. However, MPC protocols tolerating $k-1$ corruptions require computationally heavy public-key encryption techniques and are not very efficient. In this work we will therefore work with a slightly weaker notion of privacy: the information gained by any *minority* of the servers is differentially private. This allows the MPC solution to be much more efficient and to achieve unconditional, information-theoretic security requiring no computational assumptions – this makes our protocol immediately secure even against the threats of quantum computing.

A popular approach to differentially private protocols in distributed settings is the *shuffle model* [BEM+17, CSU+19b] in which scalable techniques from cryptography are combined with techniques from differential privacy, often allowing utility close to what is possible in the central model. However, existing protocols for selection use private summation, which is known to require much more noise than selection. It is likely that there is a fundamental obstacle to achieving better utility for selection in the shuffle model, due to the lower bound of [CU21] which holds for a wide class of mechanisms in the shuffle model. Another general tool for distributed differential privacy, *secure aggregation* [GX17], faces the same problem, namely that the magnitude of noise needs to grow polynomially with the dimension $d$ of the input vectors. Finally, we mention *local differential privacy* (LDP) [DJW13], in which each input vector is independently made differentially private, and where

the magnitude of noise grows polynomially in the number $n$ of input vectors.

Given that existing distributed methods for the selection problem are far from matching what is possible in the central model, and since we know that *in principle* it is possible to simulate the central model with MPC techniques, Steinke [Ste20] suggests to solve selection via an MPC implementation of argmax on secret-shared sums, but states that further investigation about the practicality is needed. In this work we perform such an investigation, modifying the approach in several ways to achieve the best fit with scalable MPC techniques. The contributions of this work are as follows:

- We present the Noise-and-round mechanism (Section 3), a distributed differentially private selection algorithm with utility guarantees close to the best algorithms in the central model.
- We introduce the first demonstration of the multi-central model for the selection problem using MPC techniques (Section 4).
- We design a new combination of integer secret sharing and existing MPC techniques which is tailored to perform a secure and efficient distributed computation of differentially private selection. In particular, this allows non-interactive truncation of input data so that approximate comparisons can be performed more efficiently than previously known.
- We provide an empirical evaluation of the utility and scalability of Noise-and-round using both synthetic and real-world data for the 3-servers case (Section 5).

## 2 TECHNICAL OVERVIEW

*Problem formulation.* The selection problem is perhaps the simplest instance of "heavy hitters," a problem ubiquitous in data analysis and machine learning. Given a collection of vectors $x^{(1)}, \ldots, x^{(n)} \in \{0, 1\}^d$ it asks to report the index of an "approximately largest" entry in $x = \sum_{j=1}^{n} x^{(j)}$. More precisely, the task is to report an index $i$ such that $x_i \geq \max_\ell(x_\ell) - \alpha n$, where $\alpha \in (0, 1)$ is an approximation parameter specifying the (additive) error within which $x_i$ is largest. This problem is a special case of general heavy hitters problems, which asks for the most frequently occurring elements in a multiset.

*Differential privacy.* Differential privacy [DMNS06] formalizes the worst-case information leakage of any output from an algorithm. Given two neighboring datasets as input differential privacy limits how much the output distributions can differ. We say that a pair of datasets are neighboring, denoted $x \sim x'$, if and only if $x$ and $x'$ differ on exactly one element. In this paper, we work in the bounded setting where the dataset's size is fixed.

**Definition 2.1** ([DMNS06] $(\varepsilon, \delta)$-differential privacy). A randomized mechanism $\mathcal{M}$ satisfies $(\varepsilon, \delta)$ − differential privacy if and only if for all pairs of neighboring datasets $x \sim x'$ and all set of outputs $Z$ we have $\Pr[\mathcal{M}(x) \in Z] \leq e^\varepsilon \Pr[\mathcal{M}(x') \in Z] + \delta$. If $\mathcal{M}$ satisfies $(\varepsilon, 0)$-DP we say that it satisfies $\varepsilon$-differential privacy.

*Our Approach.* We first describe our approach in the central model and then extend to the distributed setting. The technique is rather standard, but with a couple of deviations: following [DKS+21] we use one-sided noise when computing the noisy argmax, though

---

**Algorithm 1** Noise-and-round

1: **Input:** $x^{(1)}, \ldots, x^{(n)} \in \{0, 1\}^d$
2: **Parameters:** $\varepsilon > 0, \gamma \geq 1, \Delta \geq 0$
3: sample $\eta \sim \text{Geometric}(1 - e^{-\varepsilon/2})^d$
4: $w \leftarrow \text{round}_\Delta((\sum_{j=1}^{d} x^{(j)} + \eta)/\gamma)$
5: **return** $\arg\max_i(w_i)$

---

we replace the exponential distribution with a geometric distribution that works directly in the integer domain. Second, we show that the protocol is robust to *scaling and rounding* before taking argmax, which helps the efficiency of the MPC protocol.

The bottleneck in the secure computation protocol is the comparisons required to compute argmax. For this we use state-of-the-art protocols from [EGK+20]. These must be supplied initially with correlated randomness and are constructed as protocols for dishonest majority. However, we assume $k$ servers with $t$ semi-honest corruptions where $t < k/2$. Therefore, with the help of all servers, we can preprocess the correlated randomness using the honest majority protocol from [ACD+19], after which the first $t + 1$ servers run the protocol from [EGK+20]. Finally, we let data owners supply inputs as secret shares over the integers. This allows the servers to truncate the input without interaction while introducing only a small error; then the comparisons can work over fewer bits and hence be more efficient.

We believe that the semi-honest threat model is a realistic security model in many settings. For instance, when the main issue is not that the parties fear attacks from the others, but rather that no one wants to be responsible for storing the private data (and be liable if something leaks). This is a setting which often occurs in real life, and where semi-honest security provides sufficient gurantees.

However, it is possible to upgrade our approach to be secure against malicious servers. A server would then need to commit to its secret state and prove in zero-knowledge that it did the correct computation. Using modern techniques for this, the communication complexity would be essentially the same, but the computational load would be significantly larger.

## 3 ALGORITHM IN THE CENTRAL MODEL

In this section we analyze Algorithm 1, which solves selection in the central model and is well-suited for being extended to an efficient secure multi-party computation protocol (described in Section 4). The algorithm is a variant of the well-known "report noisy argmax" approach to selection, which has been proposed as a candidate algorithm on which to base an MPC implementation [Ste20].

Compared to a plain noisy argmax approach we make two modifications that will improve efficiency of the MPC protocol: 1) Use one-sided, geometric error, and 2) allow the argmax to be based on rounded values. Rounding is controlled by a parameter $\Delta$, such that for a rational number $w$, $\text{round}_\Delta(w)$ denotes an integer value (possibly the output of a randomized algorithm) that differs from $w$ by at most $\Delta$, and for inputs $\frac{x+\eta}{\gamma}$ and $\frac{\bar{x}+\eta}{\gamma}$ with $|x - \bar{x}| \leq 1$, using the same internal randomness for both inputs, satistifies:

$$\left| \text{round}_\Delta \left( \frac{x + \eta}{\gamma} \right) - \text{round}_\Delta \left( \frac{\bar{x} + \eta}{\gamma} \right) \right| \leq 1 . \tag{1}$$

**Algorithm 2** Relaxed-noise-and-round (The "Ideal Functionality")

1: **Input:** $x^{(1)}, \dots, x^{(n)} \in \{0,1\}^d$
2: **Parameters:** $p$ (noise parameter), $c$ (bits to truncate), $k$ (number of servers), $t$ (upper bound on corrupted servers),
3: for all $j \in [k]$ sample $r^{(j)} \sim \mathrm{NB}^d(1/(k-t), p)$
4: $z \leftarrow \sum_{i \in [n]} x^{(i)} + \sum_{j \in [k]} r^{(j)}$
5: $w \leftarrow \mathrm{round}_\Delta(z/2^c)$
6: **Output:** $\arg\max_i(w_i)$
7: **Leakage:** $r^{(j)}$ for $j \in [t]$ (capturing that the corrupted parties contribution to the noise are known to the adversary.)

When applied to a vector $x$, $\mathrm{round}_\Delta(x)$ is computed by rounding independently on each coordinate. Looking ahead to the distributed implementation of the algorithm, allowing this rounding error will allow us to perform truncation using a simple and efficient method. Proof in supplementary material.

LEMMA 1. *Algorithm 1 is $\varepsilon$-differentially private.*

LEMMA 2. *Algorithm 1 has error at most $2\gamma\Delta + 4\ln(d)/\varepsilon$ with probability at least $1 - 1/d$.*

PROOF. By a union bound, $\Pr[\|\eta\|_\infty > 4\ln(d)/\varepsilon] \le d \Pr[\eta_i > 4\ln(d)/\varepsilon] < 1/d$. Let $\mathcal{M}(x)$ denote the output of Algorithm 1, where $x = \sum_{j=1}^d x^{(j)}$ is the sum of the input vectors. We want to argue that the error $|x_{\mathcal{M}(x)} - \max_\ell(x_\ell)|$ is not too large. Abbreviating $i = \mathcal{M}(x)$, $j = \arg\max_\ell(x_\ell)$, and using that entries in $\eta$ are non-negative, we have

$$\mathrm{round}_\Delta\left(\frac{x_j + \eta_j}{\gamma}\right) \le \mathrm{round}_\Delta\left(\frac{x_i + \eta_i}{\gamma}\right)$$
$$\Rightarrow \frac{x_j + \eta_j}{\gamma} - \Delta \le \frac{x_i + \eta_i}{\gamma} + \Delta$$
$$\Rightarrow x_j + \eta_j - (x_i + \eta_i) \le 2\gamma\Delta$$
$$\Rightarrow |x_{\mathcal{M}(x)} - \max_\ell(x_\ell)| \le 2\gamma\Delta + \|\eta\|_\infty \ . \ \square$$

# 4 SECURE COMPUTATION OF DIFFERENTIALLY PRIVATE SELECTION

As it is common in the MPC literature, we first describe *what* we want to achieve in the form of an idealized algorithm, as it if was executed by some trusted third party—usually referred to as the "ideal functionality". This algorithm formally captures the computation that the distributed protocol will perform, as well as what kind of information is leaked to the adversary, while hiding the details on *how* the distributed protocols achieves this result. This ideal functionality, provided in Algorithm 2, has a small deviation from Algorithm 1; in particular, it adds a larger amount of noise sampled from a negative binomial distribution (some of which is leaked). Such distributed addition of noise has been used before in similar settings [GX17]. The increased level of noise allows us to perform a very simple and efficient distributed noise generation. Moreover, the noise leaked by the functionality is used to capture the fact that, in the distributed implementation of the algorithm, up to $t$ servers might be corrupted by a semi-honest adversary. We use $[n]$ to denote the set $\{1, \dots, n\}$.

LEMMA 3. *Algorithm 2 with $p = 1 - e^{-\varepsilon/2}$ and $\gamma = 2^c$ is $\varepsilon$-differentially private, even if the leakage is considered part of the output. It has error at most $2\gamma\Delta + 16\ln(d)/\varepsilon$ with probability at least $1 - 2/d$.*

PROOF. By symmetry we can assume that the leakage consists of the noise added by the first $t$ parties, i.e., $r^{(j)}$ for $j \in [t]$. Consider any fixed value of the leaked noise vectors—we will argue that the algorithm is $\varepsilon$-differentially private under the distribution induced by the remaining $k - t$ noise vectors. As before, let $x = \sum_{j=1}^d x^{(j)}$. After Line 4 we have

$$z = x + \sum_{j \in [k]} r^{(j)} = \left(x + \sum_{j \in [t]} r^{(j)}\right) + \sum_{j \in [k] \setminus [t]} r^{(j)},$$

where $\eta = \sum_{j \in [k] \setminus [t]} r^{(j)} \sim \mathrm{Geometric}(p)^d$ since it is a sum of $k-t$ negative binomials $\mathrm{NB}(\frac{1}{k-t}, p)$ (see e.g. [GX17]). Since $p = 1 - e^{-\varepsilon/2}$ this means that Algorithm 2 has the same output distribution as Algorithm 1 applied to an input with sum $\tilde{x} = x + \tilde{\eta}$, where $\tilde{\eta} = \sum_{j \in [t]} r^{(j)}$ is the additional noise added by the first $t$ parties. Since neighboring input sums $x \sim x'$ translate to neighboring input sums $\tilde{x} \sim \tilde{x}'$ we conclude that Algorithm 2 is $\varepsilon$-differentially private.

Abbreviating $i = \mathcal{M}(\tilde{x})$ and $j = \arg\max_\ell(x_\ell)$ we have, similar to the proof of Lemma 2,

$$\mathrm{round}_\Delta\left(\frac{\tilde{x}_j + \eta_j}{\gamma}\right) \le \mathrm{round}_\Delta\left(\frac{\tilde{x}_i + \eta_i}{\gamma}\right)$$
$$\Rightarrow \tilde{x}_j - \tilde{x}_i \le 2\gamma\Delta + \eta_i - \eta_j$$
$$\Rightarrow x_j - x_i \le 2\gamma\Delta + \eta_i - \eta_j - \tilde{\eta}_j + \tilde{\eta}'_i$$
$$\Rightarrow |x_{\mathcal{M}(x)} - \max_\ell(x_\ell)| \le 2\gamma\Delta + 2\|\eta\|_\infty + 2\|\tilde{\eta}\|_\infty \ .$$

Since $\|\eta\|_\infty > 4\ln(d)/\varepsilon$ and $\|\tilde{\eta}\|_\infty > 4\ln(d)/\varepsilon$ each happen with probability at most $1/d$ (the latter because the sum is dominated by a geometric distribution with parameter $p$) we are done. $\square$

## 4.1 Secret-sharing: notation and techniques

Our distributed protocol is performed by $k$-servers denoted by $S = \{S_1, \dots, S_k\}$. We assume that at most $t$ of them are corrupted by a semi-honest adversary (i.e., they follow the protocol specifications but then will try to infer more information by collecting their data) with $k = 2 \cdot t + 1$. We let $h = k - t = t + 1$ be the minimum number of guaranteed honest servers. As it is common in the secure multipary computation literature, we assume a single, monolithic adversary that controls all corrupted parties and collects all their internal states. This can be thought of as an adversary who has installed "spyware" on the corrupted servers: the adversary is able to observe everything that the servers observe, but not to change the code they are running. Finally, the servers will have slightly asymmetric roles in the protocol. The first $h$ servers are called the *computation servers*, whereas the last $t$ servers are called the *supporting servers* (note that by our assumptions on $k$ and $t$, at least one computation server is guaranteed to be honest, while we can tolerate that all the supporting servers might be dishonest).

We use an additive integer secret sharing scheme among the computing servers $S_1, S_2, \dots, S_h$. We use $[x]_\mathbb{Z}$ to denote a secret sharing of some integer $x$, consisting of shares $x_1, \dots, x_h \in \mathbb{Z}$ such that $\sum_{i=1}^h x_i = x$. For every $i \in [h]$, $S_i$ has $x_i$. In order to securely share

**Algorithm 3** Primitives for Integer Secret Sharing

1: **Addition.** $[z]_{\mathbb{Z}} \leftarrow [x]_{\mathbb{Z}} + [y]_{\mathbb{Z}}$ means that each server $S_i$ locally adds their shares, i.e., $z_i = x_i + y_i$ leading to $z = x + y$.

2: **Truncation.** $[y]_{\mathbb{Z}} \leftarrow \text{trunc}_\Delta([x]_{\mathbb{Z}}, c)$ means that each server $S_i$ locally computes $y_i = \lfloor x_i/2^c \rceil$ for all $i \in [h]$, removing the least significant $c$ bits from each share $x_i$ and rounding, leading to $x/2^c - \Delta \leq y \leq x/2^c + \Delta$, for a value $\Delta$ analyzed below.

3: **Conversion.** $[y]_{2^a} \leftarrow \text{convert}([x]_{\mathbb{Z}})$ means that each server $S_i$ locally computes $y_i = x_i \bmod 2^a$, leading to $y = x$ assuming $x \leq 2^a$ This is correct because $\sum_{i \in [h]} (x_i \bmod 2^a) \bmod 2^a = \sum_{i \in [h]} x_i \bmod 2^a = x \bmod 2^a$.

an $\ell$-bit long secret, we need that the shares are chosen uniformly at random among integers with $\ell + \kappa$ bits. This results in statistical security with negligible security error $2^{-\kappa}$ against any adversary, even if computationally unbounded. That is, the security of our distributed protocol does not rely on any computational assumption. Our distributed protocol performs additions and truncation of integer secret sharings, which are detailed in Algorithm 3.

*Truncation error.* Here we analyze $\Delta = |x/2^c - \sum_{i \in [h]} \lfloor x_i/2^c \rceil|$, the possible error incurred by truncation. The error depends on $h$, the number of shares of the secret. Consider the case of $h = 2$: if the input is secret shared among two servers, at most one carry bit may be missed when truncating the lower order bits. To generalize to larger $h$, first observe that division and rounding incurs an error of at most $e_i = |x_i/2^c - \lfloor x_i/2^c \rceil| \leq 1/2$. For shared integer $x$ and shares $x_1, \ldots, x_h$, when we divide $x/2^c$, we can write the result as $x_1/2^c + x_2/2^c + \cdots + x_h/2^c$. Then we can formulate the total error $\Delta = |\sum_{i \in [h]} x_i/2^c - \lfloor x_i/2^c \rceil| \leq \sum_{i \in [h]} e_i \leq h/2$ by the triangle inequality and then applying our bound for $e_i$. Notice that $\text{trunc}_\Delta([x]_{\mathbb{Z}}, c)$ exactly implements $\text{round}_\Delta(x/2^c)$ with $\Delta = h/2$.

## 4.2 A secure and differentially private distributed protocol for selection

We are finally ready to describe, in Algorithm 4, a secure distributed implementation of the differentially private mechanism from Algorithm 2 (the "ideal functionality"). The protocol proceeds as follows: In Line 4, all servers (computing and supporting) locally sample noise according to the negative binomial distribution, with parameter inversely proportional to the number of honest parties. The supporting servers need now to share their noise contribution to the computing servers in Line 5 (this can be done assuming using shares of size $\kappa + \log(n)$ assuming $\log(n)$ as an upper bound on the noise magnitude). This assumption is reasonable, since the bound holds with high probability based on tail bound analysis. If the sampled noise were to exceed the bound the protocol can, for example, report that the computation failed without compromising privacy. Alternatively, we can add this small probability to the differential privacy parameter delta. In Line 6 the computing servers exploit the linear nature of the secret sharing scheme to locally aggregate the input vectors and all noise contributions, in secret shared form. To do so, they each add all input shares and noise shares received from the supporting parties, as well as their own randomly generated noise. To increase efficiency the result is then truncated in Line 7,

**Algorithm 4** Distributed-noise-and-round (The MPC Protocol)

1: **Input:** Integer secret-sharings $\left[ \boldsymbol{x}^{(1)} \right]_{\mathbb{Z}}, \ldots, \left[ \boldsymbol{x}^{(n)} \right]_{\mathbb{Z}}$ representing values in $\{0, 1\}^d$

2: **Parameters:** $p$ (noise parameter), $c$ (bits to truncate), $k$ (number of servers), $t$ (upper bound on corrupted servers), $\kappa$ (security parameter used in integer secret sharing), $a = \log(n) - c + 1$ (bits for modular secret sharing)

3: $[\text{corr}]_{2^a} \leftarrow \text{preprocessing}(S_1, \ldots, S_k)$

4: $\forall j \in [k]$, $S_j$ samples $\boldsymbol{r}^{(j)} \sim \text{NB}^d(1/(k-t), p)$

5: $\forall j \in [t+2, k]$, $S_j$ secret-shares $\boldsymbol{r}^{(j)}$ as $\left[ \boldsymbol{r}^{(j)} \right]_{\mathbb{Z}}$ and send the corresponding shares to $S_1, \ldots, S_h$.

6: $S_1, \ldots, S_h$ evaluate $[\boldsymbol{z}]_{\mathbb{Z}} = \left[ \sum_{i \in [n]} \boldsymbol{x}^{(i)} + \sum_{j \in [k]} \boldsymbol{r}^{(j)} \right]_{\mathbb{Z}}$.

7: $S_1, \ldots, S_h$ compute $[\boldsymbol{y}]_{\mathbb{Z}} = \text{trunc}_\Delta([\boldsymbol{z}]_{\mathbb{Z}}, c)$

8: $S_1, \ldots, S_h$ convert $[\boldsymbol{y}]_{2^a} \leftarrow \text{convert}([\boldsymbol{y}]_{\mathbb{Z}})$

9: $S_1, \ldots, S_h$ execute $[o]_{2^a} \leftarrow \text{ArgMax}([\boldsymbol{y}]_{2^a}, [\text{corr}]_{2^a})$.

10: **Output:** Open and output $o = \text{argmax}_{j \in [d]} [\boldsymbol{y}]_{2^a}$

by removing the lowest $c$ bits (essentially dividing every value by $2^c$). The secret-sharing are then converted from integer to modular form in Line 8, to be compatible with the the secure ArgMax protocol which is invoked in Line 9. This protocol consumes correlated randomness which is generated by all servers during a preprocessing phase in Line 4. More details on how the ArgMax protocol and its preprocessing are implemented are given in Section 4.3.

*Correctness.* We argue that the output of Algorithm 4 has the same distribution as the one in the ideal functionality specified in Algorithm 2. First, note that the inputs are a secret shared version of the same inputs for the ideal functionality. In Line 4, noise is drawn according to the same distribution specified in Line 3 of Algorithm 2. Secret sharing and addition performed in Lines 5 and 6 correctly add the input values and random samples. In Line 7 we truncate using the secret shared version of trunc with the same output in secret shared form, and in Line 8 the conversion to secret sharing over a ring from Algorithm 3 is applied, and $a$ is chosen to be of appropriate size for this conversion to be lossless. Lastly, correctness of the ArgMax protocol used in Line 9 guarantees that the algorithm outputs the correct argmax value.

*Security.* Intuitively, security of the distributed protocol follows from the fact that the entire computation is performed over secret-shared values and that all employed sub-protocols are secure. More precisely, as it is common in the MPC literature, we can prove that the protocol is secure by providing a simulator that, given access to the input/output of the ideal functionality (including the leakage) simulates the view of the corrupted servers in the execution of the protocol. In our case the simulator, which takes as input the set of corrupted servers, and their inputs/outputs, will simulate the view of the corrupted servers essentially by running an execution of the real protocol but where the shares of all the honest parties are set to some dummy value (e.g., 0). The view of the corrupted servers contains all their shares and all the messages that they receive from the honest servers. This includes the messages that they receive from the honest servers in the preprocessing phase which, by

assumption on the security of the `preprocessing` protocol, can be efficiently simulated. The view contains also the shares of the noise generated by honest supporting servers in Line 4 which can be simulated (with statistical security $2^{-\kappa}$) by picking uniform random shares of the same size $\log(n)+\kappa$ bits as in the protocol. The Lines 6-8 only consist of local computation and can therefore be trivially simulated. Note however that, due to the local addition of the noise by the computing servers, the shares of $[\boldsymbol{y}]_{2^a}$ at the end of Line 8 might not be uniformly random. This does not matter, since the shares are never revealed but instead used as input in the secure `ArgMax` sub-protocol, which secure as shown in [EGK⁺20] (and in particular, internally, only reveals results of secure comparison protocols). Overall, the protocol in Algorithm 4 can be efficiently simulated with statistical error $2^{-\kappa}$ (due to the statistical security of the integer secret-sharing scheme) having access to the ideal functionality specified in Algorithm 2. This leads to the following:

**Corollary 4.1.** *Algorithm 4 with* $p = 1 - e^{-\varepsilon/2}$ *is* $(\varepsilon, 2^{-\kappa})$-*differentially private in the view of an adversary that semi-honestly corrupts any t servers. It has the same error as Algorithm 2.*

## 4.3 Details on the `ArgMax` protocol and preprocessing

There are multiple possible approaches for computing the exact argmax within an MPC protocol. We choose the state-of-the-art solution, which is to use a tree data structure, where the maximum of two values is compared to the maximum of two other values in each step. This approach requires $O(d)$ comparisons when finding the argmax of $d$ values. To perform the comparisons we use in turn the integer comparison protocol of [EGK⁺20], which requires that the parties hold som correlated random variables generated in the precomputation phase.

We proceed now to describe the necessary correlated randomness to execute the comparison from [EGK⁺20], and how to generate it: we let all $k$ servers collaborate in producing the correlated randomness. This allows us to achieve unconditional security (thanks to the honest majority assumption) but also to achieve high efficiency using the the protocol from [ACD⁺19]. This protocol allows us to perform MPC over $\mathbb{Z}_{2^a}$. In a nutshell, their idea is to consider a so called Galois extension $R$ of $\mathbb{Z}_{2^a}$. In the ring $R$ we can do Shamir-style secret sharing (of values in $\mathbb{Z}_{2^a}$) and follow the standard blueprint for honest majority MPC, to perform secure addition and multiplication. This implies an overhead factor $\log_2(k)$, which is necessary as Shamir-style secret sharing cannot be done over $\mathbb{Z}_{2^a}$ directly.

The correlated randomness needed by the protocol from [EGK⁺20] consists of additively shared random numbers modulo $2^a$, together with the bits in these numbers, also in shared form. Concretely, this means that the shares add modulo $a$ to the secret in question. Clearly, if we can create shared random bits $[b_0]_{2^a}, ..., [b_{2a-1}]_{2^a}$, this would be sufficient. Namely, if we let $r$ be the number with binary expansion $b_0, b_1, ..., b_{2a-1}$, then using only local computation we can construct

$$[r]_{2^a} = \sum_{i=0}^{a-1} 2^i \cdot [b_i]_{2^a} \ .$$

In order to get a random shared bit, we can use a trick suggested in [DEF⁺19]. It was shown there how to generate a random shared bit using secure arithmetic modulo a 2-power, at the cost of a constant number of secure multiplications. Using their algorithm, and the protocol from [ACD⁺19] to do the secure arithmetic, we can generate a sharing $[c]_R$, where $c$ is the random bit and $[\cdot]_R$ refers to the secret-sharing scheme from [ACD⁺19][1]

Finally, $[c]_R$ can be converted to $[c]_{2^a}$ using only local computation. Namely, if we let $\lambda_1, ..., \lambda_h$ be the Lagrange coefficients one would use to reconstruct a secret over $R$, and $s_1, ..., s_h$ be the shares of $c$ held by the first $h$ servers, we would have $c = \sum_{i=1}^{h} \lambda_i s_i$. So we can think of the $\lambda_i s_i$-values as additive shares of $c$. Each such share is an element from $R$, but it can be represented as a vector of $\log_2(k)$ numbers from $\mathbb{Z}_{2^a}$. Since addition in $R$ is component-wise addition, it turns out that each server can keep only one number from its additive share, discard the rest, and the result will be $[c]_{2^a}$.

To conclude, note that all three protocols in [DEF⁺19], [EGK⁺20] and [ACD⁺19] were originally presented for the malicious security setting, but since we deal with semi-honest corruptions their protocol can be greatly simplified in our setting.

*The 3 servers case.* We note that our protocol can be highly simplified in the case of $k = 3$. Under the assumption of honest majority this gives $h = 2$ and $t = 1$. Thus we have 2 computing servers and a single supporting server. This means that in Line 4 of the protocol we can simply have the supporting server act as a "dealer" and produce the correlated randomness locally, and then secret share it among the computation servers, instead of having to run a secure protocol among all 3 servers to generate the correlated randomness. This still guarantees security since if the dealer is corrupted then both of the computing servers must be honest (by assumption on $t \leq 1$).

## 5 EMPIRICAL EVALUATION

Inspired by the evaluation of the state-of-the-art differentially private selection algorithm Permute-and-flip [MS20], we run our benchmarks on the real-world data from DPBench [HMM⁺16]. Specifically, we use the same five representative datasets (Table 1, full table in Appendix A.4), and discretize them to $d = 1024$ as in [MS20]. To show the scalability of our MPC protocol, we also benchmark performance using synthetic data.

*Utility.* We implement and run our utility benchmarks using Python 3.11.3, measuring error for 1000 runs as the absolute difference between the true argmax value, and the one chosen by the algorithm. As there are no direct comparisons of differentially private algorithms that use the same trust model (the multi-central model), we compare to differentially private algorithms from both the central model (with stronger trust assumptions), and the local model (with weaker trust assumptions). Representing the best known error for the centralized model, we show Permute-and-flip, as well as the Exponential mechanism [MT07]. For the local model, we compare to bitwise Randomized response [War65], as used in RAPPOR [EPK14]. As a worst case comparison we also show the error of uniformly at random reporting an index as argmax. Lastly,

---

[1]An different preprocessing, suggested in [EGK⁺20], is less efficient, as it requires a super-constant number of secure multiplications per bit.

we show the error of using MPC to compute argmax, without guaranteeing differential privacy, via the use of Secure aggregation [GX17].

In Figure 1, we highlight the error by varying $\varepsilon$ and $r$ on three of the datasets from DPBench, for all datasets see Appendix A.4. We expect Noise-and-round to perform similar to the centralized algorithms (Permute-and-flip, Exponential mechanism) due to a low value of $k$ ($k = 3$), and better than the local algorithm (Randomized respone) and a purely random choice. As we can see, Noise-and-round performs similar to Permute-and-flip, and better than the Exponential mechanism. When $\varepsilon$ increases, error decreases and subsequently reaches 0 (note that the line disappears because of the log-scale). Interestingly, low values for $\varepsilon$ cause Randomized respone and Secure aggregation aggregation to perform similar to the completely random choice.

Additionally, we further show the impact of varying the remaining bits $r$ on the different datasets in Figure 2. The results show as expected that the effect of rounding is data dependent. HEPTH produces accurate results even when dropping a significant amount of bits, e.g., $r \geq 2$ (dropping 9 bits or more) gives similar accuracy in low privacy regimes (notice a change starting at $\varepsilon = 0.18$) as no rounding. SEARCHLOGS achieves similar accuracy for $r \geq 4$ (dropping 10 bits or less) and no rounding at all, and PATENT has a similar behavior for $r \geq 6$ (also dropping 10 bits or less). These results indicate that rounding can indeed be used to save communication overhead of the MPC protocol, while still maintaining accurate results.

*Runtime and communication.* The bottleneck for MPC in both time and communication lies in the computation of argmax using comparison operations, so we benchmark this part of the protocol. All benchmarks were carried out on AWS t3.xlarge instances, using MP-SPDZ [Kel20] to implement the protocol in the 3 servers case. In our experiments we vary the input dimensions ($d$), and the remaining bits ($r$). We report our results including preprocessing such as multiplication triples, and all time measurements reported are the average of ten executions for the same computation.

For each of three datasets from DPBench, we report the maximum value in each dataset, the number of bits necessary to represent integers in this range, as well as the runtimes and data sent in Table 1. Notice that while communication scales linearly in the number of bits necessary to represent the data, the time necessary for the evaluations are very close, and the variance in measurements is quite high. The last row in the table reports the necessary time and data necessary when truncating every entry in the dataset to 5 bits using our approach. Note that, due to security of MPC protocols, the runtime of the protocol cannot depend on the actual values that are being computed upon, but only their size. Therefore, the benchmarks after truncation are agnostic of which dataset we start from. The time and communication reported in the last line of the table correspond to the utility reported for $r = 5$ in Figure 1, and the utility of the approach without truncation is reported as well. We observe that by truncating values, the time and communication necessary for these comparisons is significantly reduced. Practitioners may choose how many bits to truncate based on their utility and time requirements, as well as the available computational resources.

Table 1: Benchmark results for given input

| Dataset | Max value ($n$) | # Bits | Results | |
|---|---|---|---|---|
| | | | Time (s) | Data sent (MB) |
| PATENT | 59602 | 16 | 1.74, std=0.12 | 2.97 |
| SEARCHLOGS | 11160 | 14 | 1.81, std=0.18 | 2.70 |
| HEPTH | 1571 | 11 | 1.73, std=0.07 | 2.29 |
| Truncated, $\alpha = 0.125$ | 31 | 5 | 1.38, std=0.20 | 1.39 |

The average time required per data point $d$ and power of two in the range $r$ is 0.15 ms, and the average communication is 0.22 kB. Note that while the communication scales linearly in $d$ and $r$, time scales linearly in $d$ but logarithmically in $r$. For the chosen range, the complexity can be approximated as linear in $r$ as well.

For synthetic data, the evaluated ring moduli $2^5, 2^{10}, 2^{15}, 2^{20}$, and $2^{25}$ could correspond either to different value ranges in a dataset before truncation or the resulting range of values after truncation. Based on experiments using synthetic data with sizes 16, 1024, 2048, 4096, and 8192, Figure 3 confirms the linear growth of necessary time and communication in $d$, as well as the logarithmic growth of time and linear growth in required communication in $r$. As expected, the savings in cost and communication by performing truncation increases with the size of the dataset and the range of values. Truncating even a few bits results in significant savings in communication and time, particularly when the dataset has several thousands of entries.

# 6 RELATED WORK

The exponential mechanism, as well as "report-noisy-max" [DR14], offer asymptotically optimal solutions to the selection problem in the central model. A mechanism with better constant factors is *permute-and-flip* introduced by [MS20]. We compare with their work by evaluating selection on the same benchmarking datasets and achieve comparable utility using the weaker trust assumptions of multi-central differential privacy.

The setting where data is not, and cannot, be gathered by a central entity, was a motivation for *local differential privacy* [DJW13], where a differentially private function of each participant's data is released. One such protocol for binary data is the classical *randomized response* protocol by Warner [War65]. We can apply randomized response to each bit of a binary vector (splitting the privacy budget), as seen for example in [EPK14], which allows us to estimate the sum of vectors with an error proportional to $\sqrt{n}$, where $n$ is the number of vectors.

Recent work [BEM+17, EFM+19, CSU+19b, Ste20] has increasingly focused on models of differential privacy that lie between the central and local models. The shuffle model [BEM+17, CSU+19b] is built on trust assumptions that are weaker than the central model,

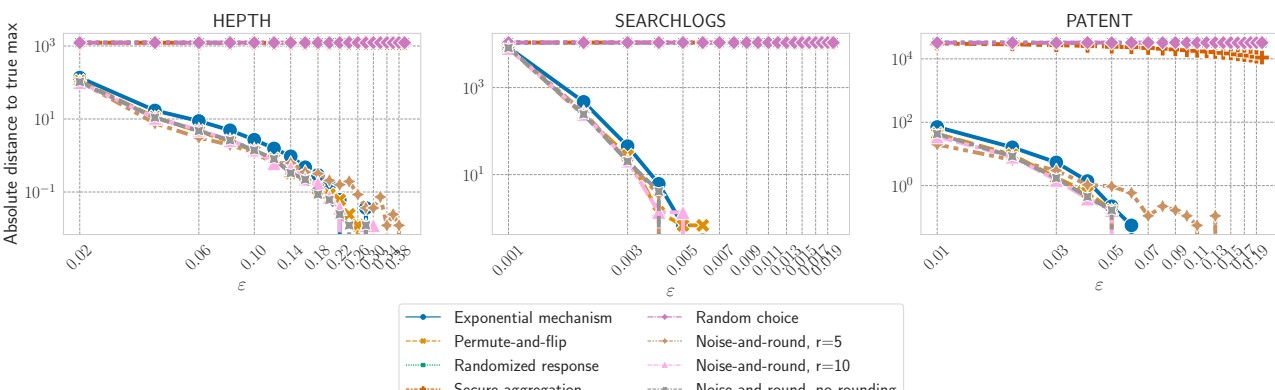

**Figure 1: Impact of $\varepsilon$ on accuracy displayed in log-log scale. Error is measured as the absolute difference between the real max, and the value of the privately chosen argmax. Lower distance is better. Notice log scale means 0 is not included, which causes some of the lines to disappear from the plot when error reaches 0.**

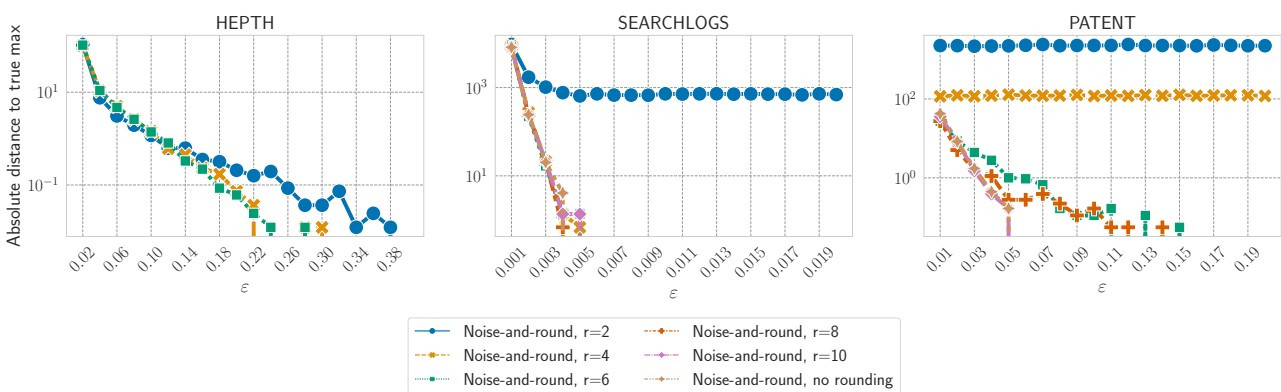

**Figure 2: Impact of rounding to $r$ remaining bits on accuracy displayed in log-log scale. Error is measured as the absolute difference between the real max, and the value of the privately chosen argmax. Lower distance is better.**

in particular a trusted shuffler, while achieving good utility for some classes of functions. However, [CU21] show an exponential separation between the central and (robust) shuffle models for the selection problem, motivating the need for alternative models.

Compared to the shuffle model, the multi-central model distributes the computation between multiple servers, as opposed to relying on the inputs being sent using an anonymous channel (e.g., using onion routing [DMS04]). [CY23] provide lower bounds for non-interactive multi-server mechanisms. The first work to consider the combination of differential privacy and MPC is [DKM+06], which focuses on distributed noise generation; however, their original work focuses on malicious adversaries, while we operate in the semi-honest security model. Some related works focus on replacing the trusted aggregator in DP with an MPC protocol for a variety of computations, while we focus on selection. [AMFD12, EKM+14, BK20] implement the exponential mechanism

with the goal of selection, yet they they perform sampling in MPC using standard techniques, a step which we avoid by allowing computing servers to sample noise locally. [BK21] focus on heavy hitters in their work. One particularly prominent application is secure aggregation [GX17, BIK+17, MPBB19, AG21], used for example in federated learning, which lends itself to the use of MPC for differentially private computations and has been implemented in practice. Secure aggregation reveals a noisy sum of inputs and requires larger error than our approach, which reveals only the output. A work closely related to ours is that of Champion, shelat and Ullman [CsU19a]: here the authors design an efficient circuit for sampling a large batch of independent coins with a given bias. As an application of their sampling technique, they provide a secure distributed implementation of the differentially private report-noisy-max mechanism. They report on an implementation for the setting of two-parties, with semi-honest security, using garbled

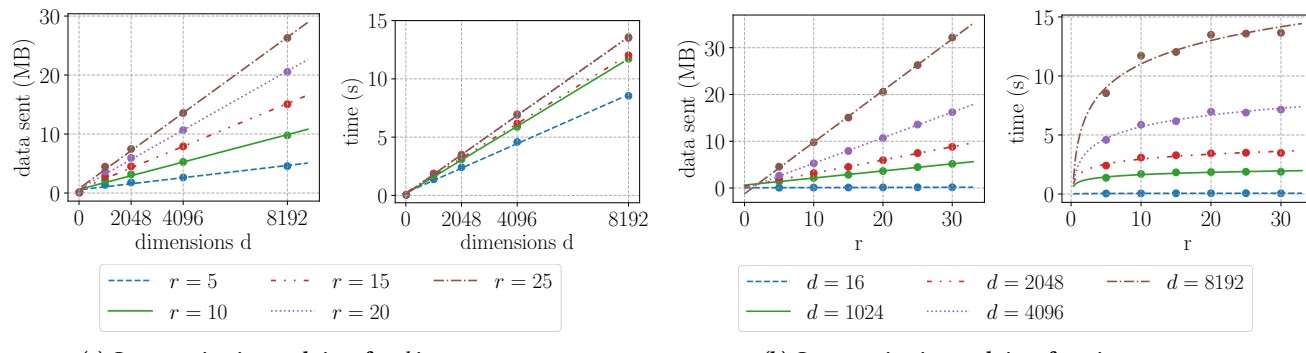

(a) Communication and time for $d$ in range 16-8192.

(b) Communication and time for $r$ in range 5-30.

**Figure 3: MPC overhead, scaling with dimensions and remaining bits; lower is better**

circuits. As the security models of the two implementations are different (3 parties in our case vs. 2 parties in their case, in both cases tolerating at most one semi-honest corruption), using different underlying technologies (secret-sharing vs. garbled circuits) making a meaningful direct comparison of the benchmarks results is somehow challenging. However, we note that our solution uses between $1 - 5\%$ of their communication (depending on our rounding factor). For instance, at $d = 8192$, their solution communicates $600MB$[2] while ours communicates between $5 - 30MB$. As both solutions scale identically with $d$, the comparison does not change at different levels of $d$. The main reason for this significant difference in bandwidth consumption is the fact that we can generate secret-shared samples from a geometric distribution without any interaction, by having the parties sample noise locally and then adding these samples to the secret-shared data. In contrast [CsU19a] performs the noise sampling by evaluating a binary circuit securely using garbled circuits. In terms of running times, the times are essentially equivalent but the comparison is made even less meaningful since the two implementations are developed on top of different MPC frameworks (Obliv-C for them and MP-SPDZ for us).

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

## A SUPPLEMENTARY MATERIAL

### A.1 Sensitivity of Rounding

Our privacy analysis will need the property (2), repeated here for convenience:

$$\left| \text{round}_\Delta \left( \frac{x + \eta}{\gamma} \right) - \text{round}_\Delta \left( \frac{\bar{x} + \eta}{\gamma} \right) \right| \leq 1 \ . \tag{2}$$

This bound follows from how the rounding function is implemeted in our MPC protocol. Note in particular that in this case we are not interested in the rounding error (i.e., the difference between the rounded value and the result of our approximate rounding function) but the sensitivity of the rounding function (i.e., the difference between the result of the approximate rounding function on two neighbouring inputs, regardless of their actual accuracy).

First remember that the users secret share $x$ to the computing servers by picking $h - 1$ uniformly random integers $x_1, \ldots, x_{h-1}$ from an appropriately large interval) and finally defining $x_h = x - \sum_{i \in [h-1]} x_i$ (resp. $\bar{x}_h = \bar{x} - \sum_{i \in [h-1]} x_i$), defining sharings $[x]_{\mathbb{Z}}$ and $[\bar{x}]_{\mathbb{Z}}$. Note that it is crucial that in this phase of the analysis we are fixing the randomness of both $\eta$ and the random shares, and we are only varying the input. Now remember that the rounding function is being implemented by having each computing server locally rounding their value which leads to

$$\text{round}_\Delta \left( \frac{x + \eta}{\gamma} \right) = \sum_{i \in [h]} \lfloor (x_i + \eta)/\gamma \rceil \ .$$

Thus we get that

$$\left| \text{round}_\Delta \left( \frac{x + \eta}{\gamma} \right) - \text{round}_\Delta \left( \frac{\bar{x} + \eta}{\gamma} \right) \right|$$

$$= \left| \sum_{i \in [h]} \lfloor (x_i + \eta)/\gamma \rceil - \sum_{i \in [h]} \lfloor (\bar{x}_i + \eta)/\gamma \rceil \right|$$

$$= |\lfloor (x_h + \eta)/\gamma \rceil - \lfloor (\bar{x}_h + \eta)/\gamma \rceil| \leq 1$$

Where the last inequality follows noticing that $x, \bar{x}$ are at most 1 apart.

### A.2 Privacy Analysis

As a warm-up we analyze an easier special case, after which we handle the general case.

LEMMA 4. *If $\Delta = 0$ and $\gamma = 1$, Algorithm 1 is $\varepsilon$-differentially private.*

PROOF. Let $\mathcal{M}(x)$ denote the output of Algorithm 1 on input with sum $x \in \mathbb{Z}^d$. Notice that $\mathcal{M}(x) = i$ if and only if

$$x_i + \eta_i \geq \max_{i' \neq i}(x_{i'} + \eta_{i'} + [i' > i]), \tag{3}$$

where $[i' > i]$ equals 1 if the condition $i' > i$ holds and 0 otherwise. Consider a neighboring dataset with sum $\bar{x}$. By definition of the neighboring relation it follows that both the left and right hand side of (3) change by at most 1 when replacing $x$ with $\bar{x}$. Using independence and the tail bound on the geometric distribution, we

**Table 2: Complexity Analysis, $k > 3$**

|         | Bits sent          | Rounds            |
|---------|--------------------|-------------------|
| Offline | $O(da^2 k \log k)$ | $O(1)$            |
| Online  | $O(akd)$           | $O(\log d \log a)$ |

**Table 3: Complexity Analysis, $k = 3$**

|         | Bits sent   | Rounds            |
|---------|-------------|-------------------|
| Offline | $O(da^2)$   | $O(1)$            |
| Online  | $O(ad)$     | $O(\log d \log a)$ |

bound

$$\Pr[\mathcal{M}(\boldsymbol{x}) = i]$$

$$= \sum_y \Pr[\max_{i' \neq i}(\boldsymbol{x}_{i'} + \boldsymbol{\eta}_{i'} + [i' > i]) = y] \Pr[\boldsymbol{x}_i + \boldsymbol{\eta}_i \geq y]$$

$$\leq \sum_y \Pr[\max_{i' \neq i}(\boldsymbol{x}_{i'} + \boldsymbol{\eta}_{i'} + [i' > i]) = y] \Pr[\boldsymbol{x}_i + \boldsymbol{\eta}_i \geq y + 2] \, e^\varepsilon$$

$$= e^\varepsilon \, \Pr[\boldsymbol{x}_i + \boldsymbol{\eta}_i \geq \max_{i' \neq i}(\boldsymbol{x}_{i'} + \boldsymbol{\eta}_{i'} + [i' > i]) + 2]$$

$$\leq e^\varepsilon \, \Pr[\bar{\boldsymbol{x}}_i + \boldsymbol{\eta}_i \geq \max_{i' \neq i}(\bar{\boldsymbol{x}}_{i'} + \boldsymbol{\eta}_{i'} + [i' > i])]$$

$$= e^\varepsilon \, \Pr[\mathcal{M}(\bar{\boldsymbol{x}}) = i] \; .$$

By symmetry we also have $\Pr[\mathcal{M}(\bar{\boldsymbol{x}}) = i] \leq e^\varepsilon \, \Pr[\mathcal{M}(\boldsymbol{x}) = i]$, as desired. □

We are ready to prove Lemma 1, which generalizes Lemma 4 to any value of the parameters:

PROOF. The key difference to the proof of Lemma 4 is that while $\boldsymbol{x}_i + \boldsymbol{\eta}_i$ and $\bar{\boldsymbol{x}}_i + \boldsymbol{\eta}_i$ differ by at most 1, we now use (2) to bound $\Pr[\mathcal{M}(\boldsymbol{x}) = i]$ by

$$\sum_y \Pr\left[\max_{i' \neq i}\left(\mathrm{round}_\Delta\left(\frac{\boldsymbol{x}_i + \boldsymbol{\eta}_i}{\gamma}\right) + [i' > i]\right) = y\right]$$

$$\Pr\left[\mathrm{round}_\Delta\left(\frac{\boldsymbol{x}_i + \boldsymbol{\eta}_i}{\gamma}\right) \geq y\right]$$

$$\leq \sum_y \Pr\left[\max_{i' \neq i}\left(\mathrm{round}_\Delta\left(\frac{\boldsymbol{x}_i + \boldsymbol{\eta}_i}{\gamma}\right) + [i' > i]\right) = y\right]$$

$$\Pr\left[\mathrm{round}_\Delta\left(\frac{\boldsymbol{x}_i + \boldsymbol{\eta}_i}{\gamma}\right) \geq y + 2\right] e^\varepsilon$$

$$= e^\varepsilon$$

$$\Pr\left[\mathrm{round}_\Delta\left(\frac{\boldsymbol{x}_i + \boldsymbol{\eta}_i}{\gamma}\right) \geq \max_{i' \neq i}\left(\mathrm{round}_\Delta\left(\frac{\boldsymbol{x}_i + \boldsymbol{\eta}_i}{\gamma}\right) + [i' > i]\right) + 2\right]$$

$$\leq e^\varepsilon \, \Pr\left[\mathrm{round}_\Delta\left(\frac{\bar{\boldsymbol{x}}_i + \boldsymbol{\eta}_i}{\gamma}\right) \geq \max_{i' \neq i}\left(\mathrm{round}_\Delta\left(\frac{\bar{\boldsymbol{x}}_i + \boldsymbol{\eta}_i}{\gamma}\right) + [i' > i]\right)\right]$$

$$= e^\varepsilon \, \Pr[\mathcal{M}(\bar{\boldsymbol{x}}) = i] \; .$$

By symmetry we also have $\Pr[\mathcal{M}(\bar{\boldsymbol{x}}) = i] \leq e^\varepsilon \, \Pr[\mathcal{M}(\boldsymbol{x}) = i]$, completing the proof. □

## A.3 MPC Protocol Analysis

We offer an analysis of the complexity associated with the operations performed by the servers in Algorithm 4, in terms of the number of necessary communication rounds and the number of bits communicated during the protocol. The local generation of noise by each server and the generation of shares of this noise by supporting servers incur no communication. However, one round of communication is necessary in order for all supporting servers to distribute their noise shares to the computing servers. Adding the shared values and the noise vectors, as well as locally truncating the resulting shares and converting them to shares over a ring, require no communication. Since the argmax is clearly the bottleneck, we will analyze that.

In order to run the ArgMax protocol, the preprocessing step involves generating additive shares modulo $2^a$ for $a(d-1)$ random bits, because each of $d-1$ comparisons requires shares of $a$ bits. Specifically in the case of 3 servers, the dealer can generate these shares locally, so only one round of communication to distribute the shares is necessary, and the number of bits to communicate will be $O(da^2)$.

Preprocessing of each secret shared bit with more than 3 servers is done using the techniques from [ACD+19]. This involves generating a random shared value and a constant number of multiplications. This can be done while communicating $O(k)$ elements of the ring over which the preprocessing is done. Due to the fact that we need "Shamir-style" secret sharing for the multiplications, we need to use a ring extension of $\mathbb{Z}_{2^a}$, where elements have size $a \log(k)$ bits, so we get communication of $O(ak \log(k))$ bits per shared random bit and so a total of $O(a^2 k \log(k))$ because we need $a$ random shared bits. Since all these bits can be created in parallel, we can do them all in a constant number of rounds. We also need $O(a)$ multiplication triples for multiplying bits, these can be done in the same complexity using the same techniques.

After precomputation is complete, running the ArgMax protocol requires $O(d)$ comparisons in a circuit structure with depth $O(\log d)$. Each comparison requires opening two secret shared values and executing two binary LT circuits. The LT circuit consists of $2a - 2$ multiplications, including two share openings each, and can be done using a circuit of depth $\log a$, where the depth indicates the number of necessary rounds. Therefore, this step incurs $O(ad)$ share openings and multiplications, and $O(\log a \log d)$ rounds of communication. Since $k$ servers are involved, these share openings and multiplications require communication $O(akd)$, which is $O(ad)$ if $k = 3$.

In total, the total communication and number of rounds when $k > 3$ is summarized in Table 2 and when $k = 3$ is summarized in Table 3.

## A.4 Utility evaluation

Here we present an individual plot for running the algorithms on each of the five datasets from DPBench. We pick the values of $\varepsilon$ to be as small as possible to capture when the most of the algorithms converge to an error of 0.

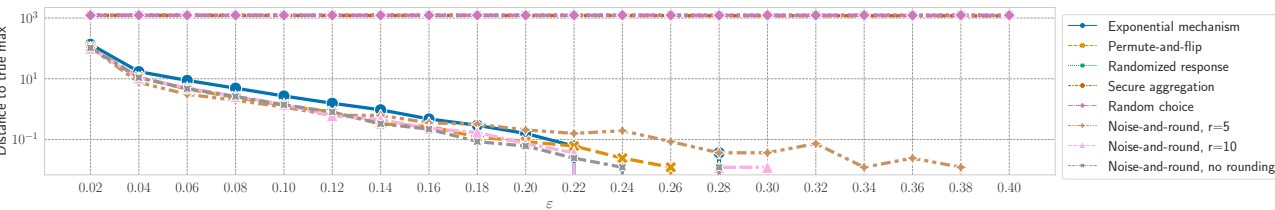

**Figure 4: HEPTH dataset. Absolute difference between the real max value, and chosen argmax on a log-log scale. Lower is better.**

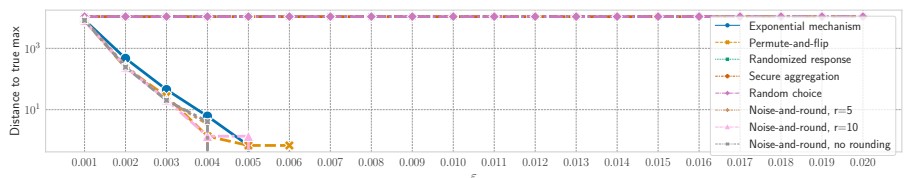

**Figure 5: SEARCHLOGS dataset. Absolute difference between the real max value, and chosen argmax on a log-log scale. Lower is better.**

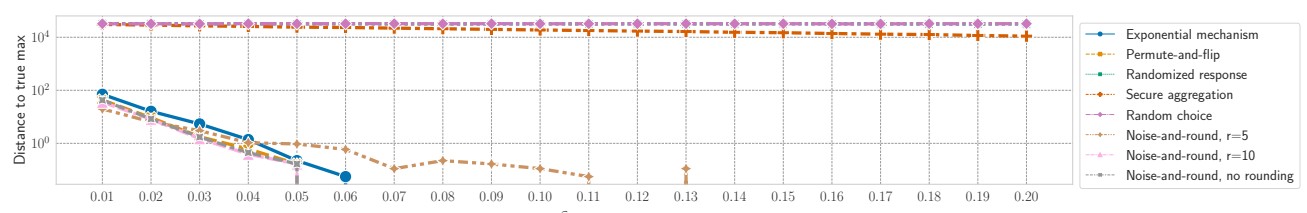

**Figure 6: PATENT dataset. Absolute difference between the real max value, and chosen argmax on a log-log scale. Lower is better.**

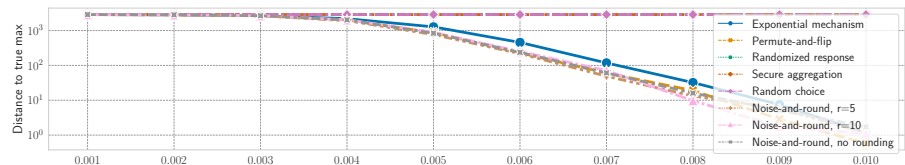

**Figure 7: MEDCOST dataset. Absolute difference between the real max value, and chosen argmax on a log-log scale. Lower is better.**



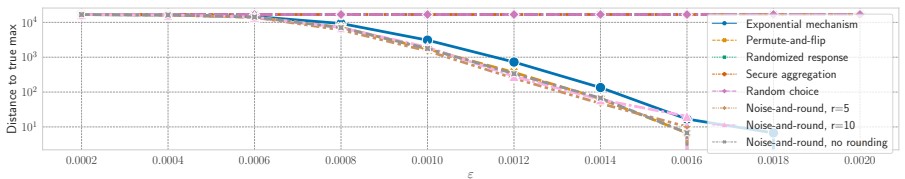

**Figure 8: ADULTFRANK dataset. Absolute difference between the real max value, and chosen argmax on a log-log scale. Lower is better.**

## A.5 Efficiency evaluation

All efficiency results for the five chosen datasets from DPBench are reported in Table 4, including the maximum value in each dataset, the number of bits necessary to represent integers in this range, as well as the runtimes and data sent. The five datasets are the same datasets chosen for evaluation by [MS20] in the Permute-and-flip mechanism: PATENT, ADULTFRANK, SEARCHLOGS, MEDCOST, and HEPTH.

**Table 4: Benchmark results for given input**

| Dataset | Max value ($n$) | # Bits | Results | |
|---|---|---|---|---|
| | | | Time (s) | Data sent (MB) |
| PATENT | 59602 | 16 | 1.74, std=0.12 | 2.97 |
| ADULTFRANK | 16836 | 15 | 1.83, std=0.15 | 2.83 |
| SEARCHLOGS | 11160 | 14 | 1.81, std=0.18 | 2.70 |
| MEDCOST | 2885 | 12 | 1.73, std=0.12 | 2.43 |
| HEPTH | 1571 | 11 | 1.73, std=0.07 | 2.29 |
| Truncated, $\alpha = 0.125$ | 31 | 5 | 1.38, std=0.20 | 1.39 |

