# OpenReview forum: "Differentially Private Selection from Secure Distributed Computing"
_ACM.org/TheWebConf/2024/Conference — TheWebConf24 Oral_

### Official Review · Reviewer_4dCi · 2023-11-23

**Novelty:** 3
**Technical Quality:** 3

**Review:**

The authors propose a novel protocol that utilizes secure multi-party computation (MPC) techniques to perform differentially private selection in distributed settings. A key feature of their approach is the ability to work with corrupted servers while maintaining privacy, addressing a significant gap in the current landscape where strong privacy guarantees often require high-trust settings. However, there are some major comments in the following.

1.	The approach of combining integer secret sharing with MPC for differentially private selection in a distributed setting is innovative. It addresses a significant challenge in distributed computing, making the study highly relevant for contemporary data privacy concerns.
2.	The Noise-and-round mechanism's ability to achieve near-central model utility in a distributed environment is noteworthy.
3.	The use of integer secret sharing in combination with existing MPC techniques seems to be well thought out and methodologically sound.
4.	The numerical experiments provide a practical perspective on the protocol's utility and scalability.

**Questions:**

1.	The empirical evaluation, though robust, is limited to a 3-server case using synthetic and real-world data. Expanding the evaluation to include a wider range of server configurations and more diverse datasets could provide a more comprehensive understanding of the protocol's performance and scalability.
2.	The use of MPC and secret sharing might be resource-intensive in terms of computation and communication overhead. This aspect could be a disadvantage in resource-constrained environments or where efficiency is a paramount concern.
3.	The format of this paper does not follow the standard, especially the format of references and citations of references.
4.	The abstract employs a narrative style that is somewhat unconventional in the context of technical and scientific publications.
5.	The paper appears to lack a dedicated conclusion section, which is a critical component of academic articles.

**Ethics Review Description:**

nil

**Reviewer Confidence:**

3: The reviewer is confident but not certain that the evaluation is correct

**Scope:**

2: The connection to the Web is incidental, e.g., use of Web data or API

---

### Official Review · Reviewer_FqUy · 2023-11-23

**Novelty:** 5
**Technical Quality:** 5

**Review:**

Overview: This work considers the problem of differentially private (DP) selection, wherein the goal is to report an approximate argmax of a sum of Boolean vectors, each held by a separate party. While this problem is fairly well-understood in the centralized DP setting, as well as the far more restrictive local DP model, this work introduces an intermediate trust regime where data is sent to $k$ servers where a majority are honest. By leveraging noising techniques from the DP literature with MPC techniques, this work obtains a DP algorithm for this setting. In the experimental evaluation, for $k=3$ servers, it is shown that the new algorithm is competitive with the centralized DP algorithms. These accuracy guarantees are supplemented with runtime investigations.

Strengths: This work introduces an interesting intermediate notion of privacy where one can obtain improved computational efficiency in practice. The theoretical analysis is complemented by extensive experimentation that suggests these algorithms are indeed competitive with centralized algorithms. In general, the paper is fairly well-written.

Weaknesses: The DP techniques themselves appear somewhat standard. Some of the discussion on the MPC analysis and implementation could be made clearer.

**Questions:**

Comments:
---Line 98: I didn't quite understand the reference to quantum computing. This may merit more elaboration.

---Line 3 of Alg 1: maybe change $d$ to $n$?

--Part of the discussion in Sections 4.2 and 4.3 are somewhat hard to follow for readers (like myself) without much background in MPC. In particular, a more formal version of the Proof of Corollary 4.1 would be useful, as would be a formal, self-contained statement of the relevant facts that are used from the existing work of [EGK+20], [ACD+19], and [DEF+19].

--The plots in Figure 1 are a little difficult to read.

--How does the runtime or experimental accuracy compare to existing locally private algorithms? Or are these not even feasible to implement?

--The $r$ parameter in the experimental evaluation appears to denote ``remaining bits,'' but it was not clear to me where this is a parameter in the pseudocode for any of the provided algorithms. But perhaps I missed where this is explained.

**Reviewer Confidence:**

1: The reviewer's evaluation is an educated guess

**Scope:**

3: The work is somewhat relevant to the Web and to the track, and is of narrow interest to a sub-community

---

### Official Review · Reviewer_9cvi · 2023-11-23

**Novelty:** 5
**Technical Quality:** 6

**Review:**

**Summary:**

The paper studies differentially private approaches for the selection problem (finding the index of the approximately largest entry in a sum of D-dimensional binary vectors). The paper focuses on a distributed setting where there are k servers, a minority <= t < k/2 of which are corrupted servers which may pool information. The main contribution is a provably private approach based on secure multi-party computation and an empirical evaluation of this approach.

The approach transforms an “idealized” algorithm into a private algorithm using secure multiparty computation. In the “idealized” algorithm, which is based on the standard approach of ReportNoisyArgmax, every party adds negative binomial noise and the output is the argmax. The secure MPC implementation splits into “computation servers” and “supporting servers” and leverages integer secret sharing to transmit the sampled noise between servers.

The paper proves that the approach is private, and empirically evaluates the approach on DPBench in comparison with several baseline approaches.


**Strengths:**
- The paper tackles an important problem of designing a practical and provably private distributed algorithm for the selection problem.
- The paper is very well-written and honestly presents its ideas in relation with related work.
- The paper explains why the approach is provably private and empirically evaluates the approach. This is a nice mix of theoretical results and empirical analysis.

**Weaknesses:**
- As the paper notes, the idea of using MPC as a distributed approach to differential privacy was already suggested by Steinke (2020). That being said, the paper concretizes this high-level idea and empirically evaluates it.
- The proof techniques are relatively straightforward and are relatively simple extensions of typical approaches in differential privacy (e.g., report argmax) and typical approaches in MPC. That being said, this is a relatively minor weakness.


**Minor comments:**
- Differentialy -> “differentially” on p.1

**Questions:**

None

**Reviewer Confidence:**

3: The reviewer is confident but not certain that the evaluation is correct

**Scope:**

3: The work is somewhat relevant to the Web and to the track, and is of narrow interest to a sub-community

---

### Official Review · Reviewer_x73t · 2023-11-24

**Novelty:** 6
**Technical Quality:** 7

**Review:**

Summary: The paper studies the selection problem. There is a set of vectors $x_1, \dots, x_n$ and the goal is to report the index $i$ such that $\sum_{j=1}^n x_i$ is highest. Specifically, the goal is to select the index $i$ under differential privacy as well as without a central trusted party. Instead, they use secret sharing to develop an algorithm implemented by $k$ servers, where $t$ servers may be corrupted or malicious. Additionally, they evaluate the algorithm on DPBench.

Strengths:
- This is a problem with clear practical implications, with clear theoretical analysis as well as empirical experiments.
- I appreciate that the authors describe the algorithm first in the central model and then in the decentralized model. This makes the paper much more accessible.
- The paper overall is very well-written and clear.

Weaknesses:
- No weaknesses to report.

**Questions:**

NA

**Reviewer Confidence:**

3: The reviewer is confident but not certain that the evaluation is correct

**Scope:**

4: The work is relevant to the Web and to the track, and is of broad interest to the community

---

### Official Review · Reviewer_WbAW · 2023-12-01

**Novelty:** 6
**Technical Quality:** 6

**Review:**

The paper focuses on the development and analysis of algorithms for differentially private selection in the context of secure distributed computing. The authors propose a novel method for selecting the maximum value from a set of data while preserving differential privacy and utilizing multi-party computation (MPC). The methodology involves adapting existing differentially private selection algorithms to work within an MPC framework, ensuring privacy and security throughout the process.
## Strengths:
- Unique approach to differentially private selection by integrating it with secure multi-party computation, addressing both privacy and security concerns effectively.
- Thoroughly evaluated algorithms on various datasets, providing a robust assessment of their performance and utility.

## Areas for Improvement:
- The complexity of implementing these algorithms in practical, real-world systems may be high, potentially limiting their accessibility and usability.

**Questions:**

- How does the performance of your algorithms compare to non-MPC-based differentially private selection methods in terms of accuracy and computational efficiency?
- How might the implementation complexity of these algorithms affect their practical adoption in real-world applications?

**Reviewer Confidence:**

2: The reviewer is willing to defend the evaluation, but it is likely that the reviewer did not understand parts of the paper

**Scope:**

3: The work is somewhat relevant to the Web and to the track, and is of narrow interest to a sub-community

---

### Decision · Program_Chairs · 2024-01-22

**Decision:**

Accept (Oral)

**Comment:**

Our decision is to accept. Please see the AC's review below and improve the work considering that and the reviewers' feedback for cemera-ready submission.

"This paper introduces a new approach to differentially private selection in secure distributed computing environments, focusing on selecting the maximum value from a dataset in a way that maintains differential privacy through multi-party computation (MPC). The method adapts existing differentially private selection algorithms to function within an MPC framework.

 Most of the referees find the approach innovative, integrating differentially private selection with MPC, which addresses both privacy and security concerns in a unique and commendable way. Moreover, the algorithms have been thoroughly evaluated on various datasets, establishing a robust assessment of their performance. The experimental results are then complemented by theoretical analysis that proves the method's competitiveness with centralized DP algorithms. Also very importantly, the problem addressed in the paper has clear practical implications, and the authors have written the narrative in a clear way, with a structure that makes the content accessible to a broader audience.

 Despite the many strengths of the paper, it has a few drawbacks. Both R1 and R5 note that the complexity of implementing these algorithms in practical systems may create a barrier to usage, limiting their accessibility and usability in real-world applications. R4 also comments that some DP techniques used in the paper seem ""standard,"" and the MPC analysis at times lacks sufficient clarity. R5 also remarks that the empirical evaluation, though robust, is somewhat limited in that it primarily focuses on a 3-server case, so expanding this to more diverse server configurations and datasets would provide a more comprehensive understanding of the protocol's performance. I would also like to see the authors discuss how the proposed algorithms compare to non-MPC-based differentially private selection methods in terms of accuracy and computational efficiency."